# The Impact of SKP2 Gene Expression in Chronic Myeloid Leukemia

**DOI:** 10.3390/genes13060948

**Published:** 2022-05-26

**Authors:** Hossam Hodeib, Dina Abd EL Hai, Mohamed A. Tawfik, Alzahraa A. Allam, Ahmed F. Selim, Mohamed E. Sarhan, Amal Selim, Nesreen M. Sabry, Wael Mansour, Amira Youssef

**Affiliations:** 1Clinical Pathology Department, Tanta University, Tanta 31527, Egypt; hossamhodeib@gmail.com (H.H.); dinaibraheem85@yahoo.com (D.A.E.H.); damirayoussef@yahoo.com (A.Y.); 2Internal Medicine Department, Tanta University, Tanta 31527, Egypt; alzahraa.allam@gmail.com (A.A.A.); a.fawzy22@yahoo.com (A.F.S.); aorta2025@yahoo.com (M.E.S.); amalibrahims@hotmail.com (A.S.); 3Clinical Oncology Department, Tanta University, Tanta 31527, Egypt; nesreensabry1eg@yahoo.com (N.M.S.); wael.mansour@med.tanta.edu.eg (W.M.)

**Keywords:** *SKP2* gene expression, imatinib, treatment response, chronic myeloid leukemia

## Abstract

Introduction: The prognosis of chronic myeloid leukemia (CML) patients has been dramatically improved with the introduction of imatinib (IM), the first tyrosine kinase inhibitor (TKI). TKI resistance is a serious problem in IM-based therapy. The human S-phase kinase-associated protein 2 (*SKP2*) gene may play an essential role in the genesis and progression of CML. Aim of the study: We try to explore the diagnostic/prognostic impact of *SKP2* gene expression to predict treatment response in first-line IM-treated CML patients at an early response stage. Patients and methods: The gene expression and protein levels of SKP2 were determined using quantitative RT-PCR and ELISA in 100 newly diagnosed CML patients and 100 healthy subjects. Results: *SKP2* gene expression and SKP2 protein levels were significantly upregulated in CML patients compared to the control group. The receiver operating characteristic (ROC) analysis for the *SKP2* gene expression level, which that differentiated the CML patients from the healthy subjects, yielded a sensitivity of 86.0% and a specificity of 82.0%, with an area under the curve (AUC) of 0.958 (*p* < 0.001). The ROC analysis for the *SKP2* gene expression level, which differentiated optimally from the warning/failure responses, yielded a sensitivity of 70.59% and a specificity of 71.21%, with an AUC of 0.815 (*p* < 0.001). Conclusion: The *SKP2* gene could be an additional diagnostic and an independent prognostic marker for predicting treatment responses in first-line IM-treated CML patients at an early time point (3 months).

## 1. Introduction

Chronic myeloid leukemia (CML) is a myeloproliferative disorder, characterized by balanced reciprocal translocation between chromosomes 9 and 22, with the subsequent generation of the BCR–ABL1 oncogenic fusion gene that encodes the chimeric BCR–ABL1 protein with enhanced tyrosine kinase activity [1,2]. The prognosis of CML patients has been dramatically ameliorated with the introduction of imatinib (IM), the first tyrosine kinase inhibitor (TKI) drug to target BCR–ABL1, thus confirming the integral role of the oncogenic fusion protein in the onset and progression of CML [3,4,5]. In the IM era, most CML patients have responded very well to IM therapy. However, a proportion of these patients either fail to respond to IM therapy (primary resistance) or become resistant after an initial response [6,7,8]. It is noteworthy that TKI resistance, either BCR–ABL1-dependent or independent resistance, is a large problem in IM-based therapy. Acquired point mutations of the ABL kinase domain, the overexpression of BCR–ABL1, or the overexpression of the multidrug resistance gene (MDR1) have been identified in the pathogenesis of BCR–ABL1-dependent/independent resistance [9,10,11,12]. To overcome IM resistance, novel TKIs have been developed and introduced into the clinic, such as the second generation of TKIs (dasatinib and nilotinib) and the more potent third-generation TKI ponatinib [13,14,15]. The initial molecular response at 3 months of TKI therapy for CML patients has been confirmed by several groups as an early response marker to predict outcomes [16,17,18,19,20]. Moreover, several studies have reported that the rate of BCR–ABL1 reduction has critical prognostic and predictive value in CML patients initially treated with TKI [21,22]. 

Ubiquitination is the process by which a ubiquitin is covalently conjugated to a protein substrate, which is carried out via a complex of enzymatic reactions, activating enzymes (E1), conjugating enzymes (E2), and ligases (E3). In addition, deubiquitinating enzymes (DUBs) are proteases that detach ubiquitin from its conjugates prior to proteolysis. In simple terms, DUBs can negatively modulate protein degradation and maintain the balance of the circulating unbound ubiquitin pool [23,24,25]. The human S-phase kinase-associated protein 2 (*SKP2*, also known as FBXL1 or p45), is an E3 ligase encoded by the *SKP2* gene [26,27]. SKP2 mostly functions as an oncoprotein, and is involved in cell proliferation, migration, invasion, angiogenesis, apoptosis, and metastasis [28]. In addition, *SKP2* overexpression was observed in several human malignancies, such as breast cancer, non-small-cell lung cancer, pancreatic cancer, gastric cancer, prostate cancer, melanoma, and hematological malignancies [29,30,31,32,33,34,35,36]. Importantly, *SKP2* has been identified in CML progression. Moreover, the inhibition of *SKP2* expression modulated TKI sensitivity in CML [37]. In addition, ubiquitin-specific peptidase 10 (USP10), a DUB, has been demonstrated as an SKP2 deubiquitinating enzyme and plays an integral role in the initiation and progression of CML, which modulates the SKP2/Bcr–Abl1 axis by stabilizing SKP2 [38]. In accordance with these findings, we postulated that *SKP2* gene expression could play a critical role in the onset and progression of CML. In the present study, we try to explore the diagnostic/prognostic impact of *SKP2* gene expression for predicting treatment responses in first-line IM-treated CML patients at an early time point (3 months), which could help to refine recommendations for treatment options at the early response stage. 

## 2. Patients and Methods

This is a prospective cohort study carried out from December 2019 to November 2021 in the Clinical Oncology Department and the Hematology/Oncology section, Internal medicine Department, Tanta University Hospital, Egypt. In total, 100 newly diagnosed CML (chronic phase) patients and 100 healthy control subjects of matched age and gender to the patients were registered in the study. The diagnosis of chronic phase (CP) was defined according to European Leukemia Net (ELN) criteria [39]. Only CML-CP patients who were ≥18 years old and had a Philadelphia+ and/or BCR–ABL1+ were included in the study. Patients who had received hydroxyurea or interferon were excluded from the analysis. Baseline risk assessment was performed using the Sokal risk score [40]. The cytogenetic and molecular analysis results were available in patients’ records. All patients were initially treated with first-line 400 mg imatinib (IM) [41], and carefully observed for a period of three months. Cytogenetic, hematologic, and molecular responses, either optimal, warning or failure, were defined according to ELN criteria [39]. With respect to BCR–ABL1 mutations, we analyzed the mutations only if resistance to IM was suspected and not carried out on a routine basis for all CML patients. BCR–ABL1 mutations were obtained from the patients’ files. All the patients were eligible for the study after approval by the Tanta University hospital ethical committee, and the study was conducted in accordance with the principles of the declaration of Helsinki; signed informed consent was obtained from all the patients and control participants enrolled in the study.

### 2.1. Peripheral Blood Collection

Whole blood was collected at the time of CML-CP diagnosis prior to the beginning of any medication and after three months of IM treatment. Whole blood was withdrawn via standard venipuncture in Vacuette Blood Collection Tubes (Greiner Bio-One, Austria) with K2EDTA for the complete blood picture and assessment of peripheral blood smear; the molecular analysis of BCR–ABL1 transcripts and measurement of SKP2 protein level was carried out via ELISA, with lithium heparin for cytogenetic analyses and sodium heparin for the separation of peripheral blood mononuclear cells (PBMCs) for RNA extraction; and cDNA synthesis and real-time reverse transcriptase–polymerase chain reaction (RT-PCR) were used for *SKP2* gene expression analysis.

### 2.2. Molecular Analysis of BCR–ABL1 Transcript

BCR–ABL1 transcript expression levels were measured through quantitative real-time PCR [17,42], using ipsogen^®^ BCR-ABL1 Mbcr IS-MMR (Cat. No. 670823, Qiagen GmbH, Hilden, Germany). ABL1 was used as the control gene. Results were expressed as BCR–ABL1 IS %. The international scale (IS) was defined as a percentage, with 100% BCR–ABL1 IS corresponding to the International Randomized Study of Interferon and STI571 (IRIS) study’s standardized baseline. The results represented on the international scale depend on either the conversion factor (CF) obtained from the reference laboratories or by using kits and reagents pre-calibrated to the World Health Organization International Genetic Reference Panel for the quantitative measurement of BCR–ABL1 mRNA [43,44,45,46,47].

### 2.3. Mononuclear Cell Isolation, RNA Extraction, and cDNA Synthesis for SKP2 Gene Expression Analysis

PBMCs were separated from heparinized whole blood through the utilization of Ficoll density gradient centrifugation using Ficoll-Paque PREMIUM (cat. no.: 17-5446-52, GE Healthcare, Germany). In brief, Ficoll-Paque PREMIUM was withdrawn using a syringe under a complete aseptic technique to a sterile tube. Blood was equally diluted with phosphate buffered saline (PBS), then carefully layered on Ficoll-Paque PREMIUM before centrifugation at 400× *g* for 30 min at room temperature. The upper plasma layer was discarded; next, the mononuclear layer was transferred to a sterile centrifuge tube. The cell isolates (mononuclear cells) were washed and suspended in 500 μL of clinical buffer. Total RNA was immediately extracted from PBMCs using the QIAamp RNA extraction blood mini kit (cat. no.: 52304, QIAGEN GmbH, Hilden, Germany). In brief, leukocyte pellets from PBMNCs were applied for the lysis and homogenization of lysate. Adding ethanol allowed total RNA binding to the QIAamp membrane. High-quality RNA was then eluted. Total RNA was stored at −80 °C until the time of the assay. NanoDrop 2000 (Thermo Fisher Scientific, Waltham, MA, USA) was used to assess the purity and integrity of total RNA. Complementary DNA (cDNA) was synthesized from RNA using the QuantiTect Reverse Transcription Kit (cat. no. 205313, Qiagen GmbH, Hilden, Germany), according to the manufacturer’s instructions.

### 2.4. Real-Time Reverse Transcriptase–Polymerase Chain Reaction (RT-PCR)

The expression levels of *SKP2* (gene of interest) and GAPDH (reference gene) were measured using quantitative RT-PCR via the QuantiTect SYBR Green PCR Kit (cat. no. 204141, Qiagen GmbH, Hilden, Germany), and PCR reaction was carried out in Applied Biosystems StepOne™Real-Time PCR Systems (Applied Biosystems, Foster City, CA, USA) at the molecular biology section, clinical pathology department, Tanta University Hospital, Egypt. In brief, a PCR reaction was performed in a final volume of 20 μL. One μL of purified cDNA was added to a volume of 19 μL of the amplification mix (9 μL of Master Mix, 0.5 μL of each of the reverse and the forward primers (*SKP2*/ GAPDH) and 9 μL of nuclease-free H2O). Thermal profile for PCR was programmed as follows: 95 °C for 15 min (initial activation), followed by 40 cycles of 94 °C for 15 s (denaturation), 55 °C for 30 s (annealing), and 72 °C for 30 s (extension). The following primers were used: 

*SKP2* forward, 5′-AGTCTCTATGGCAGACCTTAGACC-3ʹ and reverse, 5′-TTTCTGGAGATTCTTTCTGTAGCC-3ʹ, GAPDH forward, 5′-CTCCTCCTGTTCGACAGTCAG-3′ and reverse, 5′-CCCAATACGACCAAATCCGTT-3′.

All reactions were carried out in duplicate. The data were expressed as the relative expression of *SKP2* relative to GAPDH, as determined by the 2^−ΔΔCT^ method.

### 2.5. SKP2 Protein Level Assay

The SKP2 protein level was measured using the Human SKP2 ELISA Kit (cat no: LS-F33652, Lifespan Biosciences, WA, USA) (sensitivity: 0.094 ng/mL; range: 0.156–10 ng/mL). In brief, SKP2 protein level was estimated by sandwich-ELISA-based assay in which standards/samples were added to wells pre-coated with a specific capture antibody. The unbound standard/sample was washed. Next, biotin-conjugated antibody was added to allow binding to the captured antigen. The unbound antibody was washed. Next, streptavidin–horseradish peroxidase (HRP) conjugate was added to allow binding to the biotin. The unbound streptavidin–HRP conjugate was washed. Next, TMB substrate was added and the reaction with the HRP enzyme occurred. A colored product was formed in proportion to the amount of human SKP2 present in the sample/standard. The reaction was terminated by addition of sulfuric acid stop solution.

### 2.6. Statistical Analysis

Data were supplied to the computer system and analyzed by the IBM SPSS software package, version 20.0. (Armonk, NY, USA: IBM Corp). The Kolmogorov–Smirnov test was applied to identify the normality of distribution of the variables. A chi-square test was applied to compare groups for categorical variables. Student’s *t*-test was applied to compare two groups for normally distributed quantitative variables. The Mann–Whitney U test was applied to compare two groups for non-normally distributed quantitative variables. The Wilcoxon signed-ranks test was assessed for the comparison between two periods for non-normally distributed quantitative variables. The Spearman coefficient was applied to correlate between quantitative variables. The receiver operating characteristic curve (ROC) was performed to determine the diagnostic/prognostic performance of the markers: an area of more than 50% shows acceptable performance, and an area of approximately 100% is the best performance on the test. To compare different groups for quantitative abnormally distributed variables, the Kruskal–Wallis test was used. The odds ratio (OR) was used to calculate the ratio of the odds and the 95% confidence interval of an event occurring in one risk group compared to the odds of it occurring in the non-risk group. The significance of the obtained results was judged at the 5% level.

## 3. Results

In total, 100 patients recently diagnosed with CML and 100 healthy subjects (control group) were included. There was no detectable significant difference between the studied groups with respect to age or gender (*p* > 0.05). The demographic and baseline characteristic details of the studied groups are illustrated in Table 1. First, we evaluated the *SKP2* gene expression levels among the CML patients and the healthy subjects. The mean *SKP2* gene expression levels were significantly higher in the CML patients compared to the control group (2.3 ± 0.9 vs. 0.8 ± 0.3 (*p* < 0.001)) (Table 1, Figure 1). These findings pointed to the molecular pathogenic role of *SKP2* in the onset and the development of CML. Next, we investigated the SKP2 protein levels to verify both the gene-level and the protein-level expression in both the CML patients and the healthy subjects. Interestingly, the mean SKP2 protein levels were significantly higher in the CML patients compared to the control subjects (4.4 ± 1.5 vs. 1.2 ± 0.6, (*p* < 0.001) (Table 1, Figure 2). In addition, the SKP2 protein levels were positively correlated with the gene expression levels (r = 0.997, *p* < 0.001). Next, we evaluated the *SKP2* gene expression and protein levels in the CML patients initially treated with first-line IM after 3 months. Through kinetic measurement, at sequential time points from the beginning of the treatment until the early response stage, we observed that the *SKP2* gene expression and protein levels were significantly downregulated (2.3 ± 0.9 vs. 1.4 ± 0.9 and 4.4 ± 1.5 vs. 2.5 ± 1.6) (*p* < 0.001). Moreover, the BCR–ABL1 transcript significantly decreased at 3 months from the starting point of the therapy (69.4 ± 18.2 vs. 19.1 ± 24.7) (*p* < 0.001) (Table 2). Furthermore, no significant correlation was observed between the BCR–ABL1 transcript and the *SKP2* gene expression levels in the CML patients (*p* = 0.627). In addition, no significant correlation was observed between the SOKAL score and the *SKP2* gene expression levels in the CML patients (*p* = 0.861). Next, we evaluated the treatment response at the early response stage (3 months) after the administration of 400 mg IM in the newly diagnosed CML patients; 4/100 (4%), 34/100 (34%), and 29/100 (29%) failed to achieve complete hematologic response (CHR), complete cytogenetic response (CCR), and optimal molecular response (BCR–ABL1 IS% < 10), respectively. Taking this a step further, we examined the *SKP2* gene expression levels in the newly diagnosed CML patients with respect to treatment response (cytogenetic, hematologic, and molecular). Interestingly, the mean expression levels of *SKP2* were significantly higher in the patients who failed to achieve either CHR, CCR, or optimal molecular response (3.7 ± 0.1 vs. 2.2 ± 0.8, 2.9 ± 0.7 vs. 1.9 ± 0.7, 3.0 ± 0.7 vs. 2.0 ± 0.7, *p* < 0.001), respectively (Table 3). These findings suggested that elevated expression levels of the *SKP2* gene could contribute to primary treatment resistance or failure. Among the 34/100 (34%) patients who developed primary resistance or failure, 12/34 (35%) and 22/34 (65%) were identified as positive and negative for BCR–ABL1 mutations, respectively. Additionally, the mean expression levels of *SKP2* were significantly lower in the patients carrying BCR–ABL1 mutations compared to their counterparts who were negative for the mutations (2.5 ± 0.7 vs. 3.1 ± 0.7, *p* = 0.01) (Table 3). Next, we tried to explore the diagnostic impact of the *SKP2* gene; the ROC analysis revealed that the best cut-off value for the *SKP2* gene expression level that differentiated the CML patients from the healthy subjects (control group) was >1.08, yielding sensitivities of 86.0% and 82.0%, with an AUC of 0.958 (*p* < 0.001) (Figure 3), suggesting that *SKP2* gene expression could be an additional diagnostic marker for CML. Next, we tried to explore the prognostic impact of the *SKP2* gene to predict the treatment responses at the early time point (3 months); the ROC analysis revealed that the best cut-off value of the *SKP2* gene expression level that differentiated the optimal response from the warning/failure response was >2.42, yielding a sensitivity of 70.59% and a specificity of 71.21%, with an AUC of 0.815 (*p* < 0.001) (Figure 4), suggesting that *SKP2* gene expression could be an independent prognostic marker to predict the treatment responses in the CML patients at the early time point (3 months). Ultimately, we evaluated the risk prediction of the warning/failure response. Surprisingly, the patients with elevated *SKP2* expression levels had an increased risk of failure warning/failure response (OR = 5.259; 95% CI = 2.568–10.772; *p* < 0.001 and adjusted OR = 5.234; 95% CI = 2.555–10.725; *p* < 0.001, respectively, adjusted by SOKAL score) (Table 4). These findings suggested that the presence of elevated *SKP2* expression levels could be an independent risk factor for primary treatment resistance or failure. 

## 4. Discussion

In the 1960s, the Philadelphia chromosome was discovered by Nowell and Hungerford [48]; this abnormality was quickly designated and confirmed by Caspersson et al. [49]. In the 1980s, the oncogenic BCR–ABL1 was identified as the molecular cause of CML development [50]. TKIs inhibit BCR–ABL1 kinase activity and have been listed as the drug of choice in CML therapy [39]. Additionally, primary TKI resistance is also commonly observed in CML patients [7]. Notably, BCR–ABL1-dependent TKI resistance has been attributed to mutations within the BCR–ABL1 kinase domain, while BCR–ABL1-independent resistance has been attributed to the abnormal activation of a pathway or gene expression [51,52]. Currently, the main concern is to identify the underlying molecular mechanisms of IM resistance to develop a novel therapeutic target for the improvement of the chemotherapeutic effects in CML. As mentioned above, SKP2 not only functions as an oncoprotein, but has also been identified in cancer-associated drug resistance [28,53,54,55,56].

Accumulating data in the literature suggest that *SKP2* could be a potential therapeutic target in IM sensitivity/resistance in newly diagnosed CML patients. In the present study, we attempted to introduce the *SKP2* gene, a crucial molecular player in malignancy, which may play a critical role in the onset and progression of CML. The main important findings obtained from the present work were that the *SKP2* gene expression and protein levels were significantly upregulated in the CML patients compared to the control group; moreover, these levels were dramatically downregulated in the CML patients initially treated with IM at the early response stage. In addition, BCR–ABL1 transcript significantly decreased at 3 months from the starting point of the therapy. In accordance with prior reports, the *SKP2* gene expression was significantly higher in the patients with CML compared to the healthy donors. Furthermore, the inhibition of *SKP2* expression greatly enhances the sensitivity of CML cells to IM treatment [37]. Similar to our findings, USP10 and SKP2 proteins were upregulated in CML patients compared to the healthy control subjects [38]. This mild discrepancy could be attributed to the different laboratory technique (Western blot assay), which was not applied in the present study. It is interesting to note that *SKP2* overexpression was observed in multiple solid tumors as well as hematological malignancies [29,30,31,32,33,34,35,36]. Importantly, the results obtained from the present study revealed that the expression levels of *SKP2* were significantly higher in the CML patients with warning/failure responses to IM treatment compared to the CML patients with optimal responses at the early response stage. Furthermore, no significant correlation was observed between the BCR–ABL1 transcript and *SKP2* gene expression levels in the CML patients in the present study, suggesting that the *SKP2* gene might be associated with BCR–ABL1-independent resistance to IM. It is noteworthy that SKP2 mediates its effect via the K63-linked ubiquitination of BCR–ABL1, with the subsequent activation of BCR–ABL1 signaling. Moreover, the deubiquitinating activity of USP10 modulates the *SKP2* expression and, thus, the activation of BCR–ABL1 signaling. Additionally, USP10 is involved in the regulation of the cell cycle via its effect on the BCR–ABL1/SKP2/P27 axis. Overall, USP10 exerts its effect against ubiquitination and SKP2 degradation with a subsequent increased SKP2 expression level and, thus, an enhanced activation of BCR–ABL1 signaling [38]. In addition, BCR–ABL1-independent resistance to IM has been associated with transcription factor NF-κB, protein kinase C, and HDACs in CML, but not with BCR–ABL1 [57,58,59]. Interestingly, the ROC analysis of the *SKP2* gene expression suggested that *SKP2* gene expression could be an additional diagnostic marker in the context of CML and an independent prognostic marker to predict treatment responses in CML patients at an early time point. In addition, the high odds ratio (OR) of the *SKP2* gene indicated a more precise prediction of treatment responses. 

To the best of our knowledge, there are no available data in the literature ascribing the expression pattern of the *SKP2* gene to the treatment response, and the available information is suggestive of the role of *SKP2* in the pathogenesis of CML, although it remains to be elucidated. In line with earlier studies, Liu et al. [60] reported that high *SKP2* expression levels were observed in primary gastrointestinal stromal tumors (GISTs), and that these levels may pose an increased risk of disease progression. Furthermore, a fraction of primary GISTs that do not respond to IM by apoptosis are eliminated from the proliferative pool by entering quiescence via the modulation of the anaphase-promoting complex (APC)/CDH1-SKP2-p27^Kip1^ signaling axis. Moreover, Zhang et al. [61] reported that IM and GNF-5 inhibited hepatocellular carcinoma (HCC) cell growth via the downregulation of *SKP2* expression and the upregulation of both p27 and p21 levels in HepG2 cells, inducing G0/G1 phase cell-cycle arrest. In brief, our results confirmed and extended prior reports clarifying the in vitro cellular response to IM in different malignancies in general and CML in particular. The dysregulation of *SKP2* could be attributed to the multiple signaling pathways with the complexity of the underlying pathological mechanisms in human malignancies. The results obtained from the present work revealed that 34% of our patients developed primary resistance or failure, and 35% of them harbored BCR–ABL1 mutations. Similar findings were reported by Soverini et al. [62], who observed that 45/152 (30%) patients with primary resistance had BCR–ABL1 mutations in an Italian population. Moreover, Liu et al. [63] reported that 54/175 (30.9%) patients with TKI resistance carried BCR–ABL1 mutations in a Chinese population. This discrepancy could be attributed to differences in the time points of the mutational screening, the disease phase, and the sensitivities of the techniques performed. It is important to note that although these results are promising, we cannot confirm them due to the extremely low number of patients involved in the present study. Ultimately, we introduced the *SKP2* gene as an E3 ligase, orchestrating a complex pathological process involved in the onset and progression of CML. Furthermore, our study might pave the way for further research and the development of a therapeutic approach for newly diagnosed CML patients.

## 5. Conclusions

The *SKP2* gene could be an additional diagnostic marker in the context of CML and an independent prognostic marker to predict treatment response in first-line IM-treated CML patients at an early time point (3 months). Moreover, elevated *SKP2* expression levels could be an independent risk factor for primary treatment resistance or failure in newly diagnosed CML patients.

## 6. Limitations of the Study

The present study has multiple limitations that should be mentioned. First, our study included a relatively small number of CML patients. Second, the present study did not provide mechanistic or experimental evidence to elucidate the exact molecular mechanism of the *SKP2* gene (the experiment was performed in vitro). Finally, this was a single-center study related to a population with a homogeneous ethnic background.

## Figures and Tables

**Figure 1 genes-13-00948-f001:**
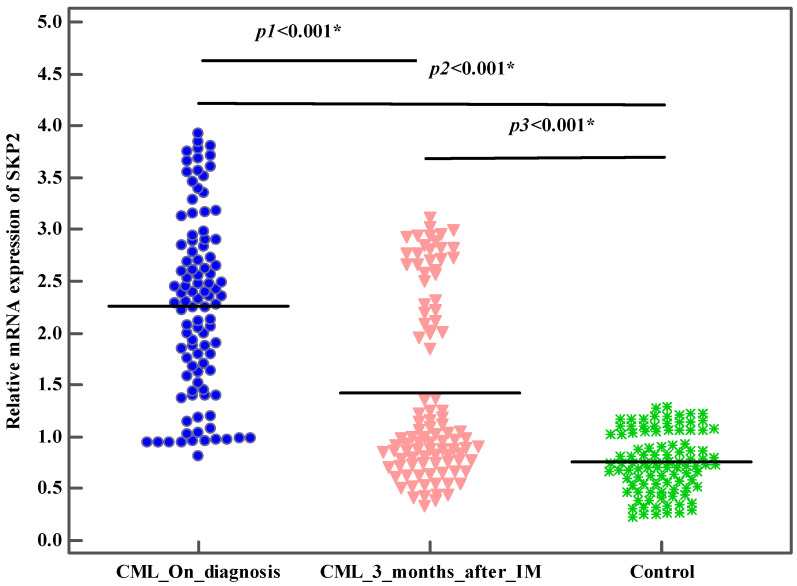
The relative mRNA expression levels of *SKP2* at the time of diagnosis (*n* = 100), at 3 months after IM in CML patients (*n* = 100) and the control subjects (*n* = 100). The blue symbols represent the relative mRNA expression levels of *SKP2* gene at the time of diagnosis; the pink symbols represent the relative mRNA expression levels of *SKP2* gene 3 months after IM; and the green symbols represent the relative mRNA expression levels of *SKP2* gene of the control subjects. *p_1_*: *p* value for **Wilcoxon signed-ranks test** for comparison between the relative mRNA expression levels of *SKP2* gene of CML patients at the time of diagnosis and 3 months after IM. *p_2_*: *p* value for **Mann–Whitney test** for comparison between the relative mRNA expression levels of *SKP2* gene at the time of diagnosis and the control subjects. *p_3_*: *p* value for **Mann–Whitney test** for comparison between the relative mRNA expression levels of *SKP2* gene 3 months after IM and the control subjects. *: statistically significant at *p* ≤ 0.05.

**Figure 2 genes-13-00948-f002:**
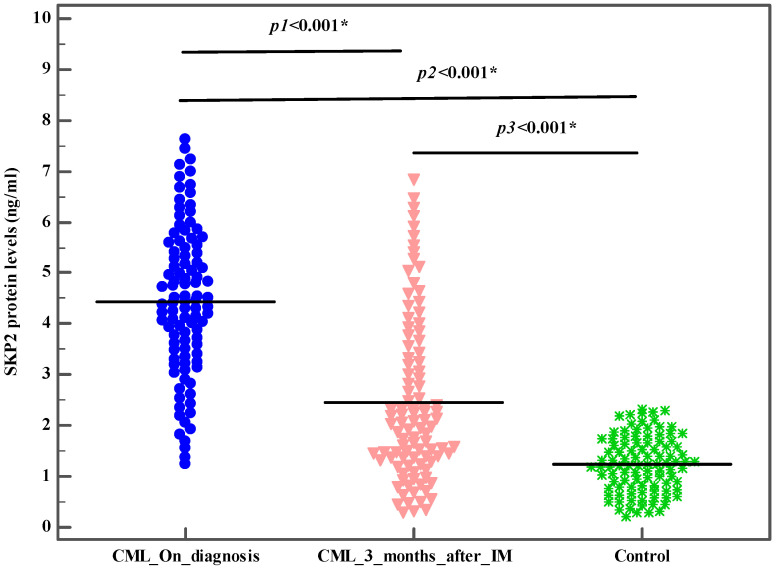
SKP2 protein levels (ELISA) at the time of diagnosis, at 3 months after IM in CML patients (*n* = 100) and the control subjects (*n* = 100). The blue symbols represent the SKP2 protein levels at the time of diagnosis; the pink symbols represent SKP2 protein levels 3 months after IM; and the green symbols represent SKP2 protein levels of the control subjects. *p_1_*: *p* value for **Wilcoxon signed-ranks test** for comparison between the SKP2 protein levels of CML patients at the time of diagnosis and 3 months after IM. *p_2_*: *p* value for **Mann–Whitney test** for comparison between the SKP2 protein levels at the time of diagnosis and the control subjects. *p_3_*: *p* value for **Mann–Whitney test** for comparison between the SKP2 protein levels 3 months after IM and the control subjects. *: statistically significant at *p* ≤ 0.05.

**Figure 3 genes-13-00948-f003:**
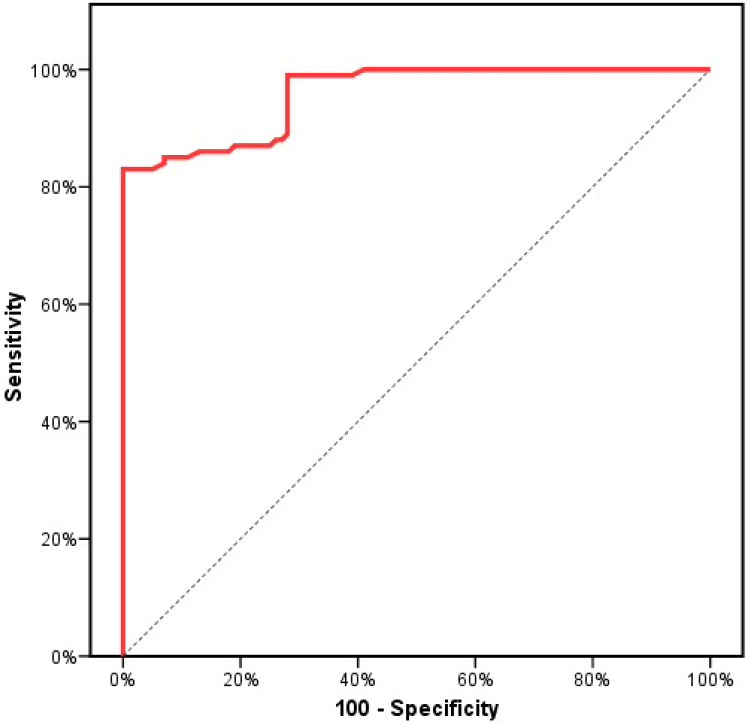
ROC curve for relative expression of *SKP2* to discriminate CML patients (n = 100) from the control subjects (*n* = 100).

**Figure 4 genes-13-00948-f004:**
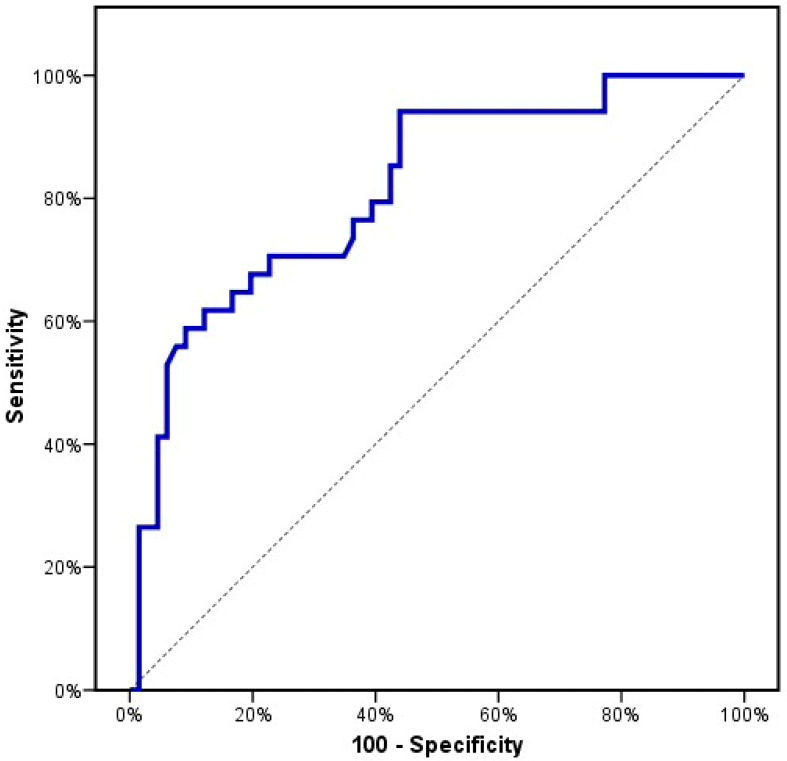
ROC curve for relative expression of *SKP2* at the time of diagnosis to predict treatment response (discriminate optimal response from warning/failure response).

**Table 1 genes-13-00948-t001:** Demographic and baseline characteristic data of the two studied groups.

	CML (*n* = 100)	Control (*n* = 100)	Test of Sig.	*p*
**Age (years)**				
Mean ± SD.	54.4 ± 7.7	52.9 ± 7.9	T = 1.396	0.164
Median (Min.–Max.)	58 (39–63)	55.5 (39–64)		
**Gender**				
Male	73 (73.0%)	70 (70.0%)	χ^2^ = 0.221	0.638
Female	27 (27.0%)	30 (30.0%)		
**BCR–ABL1 IS%**				
Mean ± SD.	69.4 ± 18.2	NA		
Median (Min.–Max.)	69.5 (33–108)	NA	NA	NA
**SOKAL**				
Low	32 (32%)	NA	NA	NA
Intermediate	41 (41%)	NA		
High	27 (27%)	NA		
**Relative expression of *SKP2***				
Mean ± SD.	2.3 ± 0.9	0.8 ± 0.3		
Median (Min.–Max.)	2.3 (0.8–3.9)	0.7 (0.2–1.3)	U = 416.50 *	<0.001 *
**Protein levels of SKP2** **(ELISA) (ng/mL)**				
Mean ± SD.	4.4 ± 1.5	1.2 ± 0.6	U = 193.0 *	<0.001 *
Median (Min.–Max.)	4.4 (1.2–7.6)	1.2 (0.2–2.3)		
**BCR–ABL1 Mutations**				
NA	66 (66%)	NA		
Yes	12 (12%)	NA		
No	22 (22%)	NA		

SD: standard deviation; t: Student’s *t*-test; χ^2^: chi-square test; U:Mann–Whitney test; *p*: *p* value for comparison between the studied groups; *: statistically significant at *p* ≤ 0.05; NA: not analyzed.

**Table 2 genes-13-00948-t002:** BCR–ABL1 IS%, *SKP2* gene expression levels and SKP2 protein levels at the time of diagnosis and at 3 months after IM in CML patients (*n* = 100).

	On Diagnosis	3 Months after IM	Z	*p*
**BCR** **–** **ABL1 IS%**				
Mean ± SD.	69.4 ± 18.2	19.1 ± 24.7	8.607 *	<0.001 *
Median (Min.–Max.)	69.5 (33–108)	7 (1–90)		
**Relative expression of *SKP2***				
Mean ± SD.	2.3 ± 0.9	1.4 ± 0.9	8.110 *	<0.001 *
Median (Min.–Max.)	2.3 (0.8–3.9)	1 (0.3–3.1)		
**Protein levels of SKP2** **(ELISA) (ng/mL)**				
Mean ± SD.	4.4 ± 1.5	2.5 ± 1.6	8.555 *	<0.001 *
Median (Min.–Max.)	4.4 (1.2–7.6)	2 (0.3–6.8)		

SD: standard deviation; Z: Wilcoxon signed-ranks test; *p*: *p* value for comparison between on diagnosis and 3 months after IM; *: statistically significant at *p* ≤ 0.05.

**Table 3 genes-13-00948-t003:** *SKP2* gene expression levels (at time of diagnosis) and the treatment response (at 3 months after IM) in CML patients (*n* = 100) and BCR–ABL1 mutations (*n* = 34).

	N	*SKP2* On Diagnosis		Test of Sig.	*p*
		Mean ± SD	Median (Min.–Max.)		
**Hematologic Response**					
Yes	96	2.2 ± 0.8	2.3 (0.8–3.9)	U = 18.0 *	<0.001 *
No	4	3.7 ± 0.1	3.7 (3.6–3.8)		
**Cytogenetic Response**					
Yes	66	1.9 ± 0.7	1.9 (0.8–3.9)	U = 416.0 *	<0.001 *
No	34	2.9 ± 0.7	3 (1.2–3.8)		
**Molecular Response**					
BCR–ABL1 IS% < 10	71	2.0 ± 0.7	1.9 (0.8–3.9)	U = 331.0 *	<0.001 *
BCR–ABL1 IS% ≥ 10	29	3.0 ± 0.7	3.2 (1.2–3.8)		
**Treatment Response**					
Optimal	66	1.9 ± 0.7	1.9 (0.8–3.9)	U = 416.0 *	<0.001 *
Warning/Failure	34	2.9 ± 0.7	3 (1.2–3.8)		
**BCR–ABL1 Mutations**					
Yes	12	2.5 ± 0.7	2.3 (1.2–3.7)	U = 61.0 *	0.010 *
No	22	3.1 ± 0.7	3.2 (1.2–3.8)		

SD: standard deviation; U: Mann–Whitney test; *p*: *p* value for comparison between the studied groups; *: statistically significant at *p* ≤ 0.05.

**Table 4 genes-13-00948-t004:** Risk prediction of treatment response (warning/failure response).

	Crude Odds Ratio		Adjust Odds Ratio #	
	*p*	OR (95% CI)	*p*	OR (95% CI)
* **SKP2** * **on Diagnosis**	<0.001 *	5.259 (2.568–10.772)	<0.001 *	5.234(2.555–10.725)

OR: odds ratio; CI: confidence interval; #: odds ratio adjusted by SOKAL score; *p*: *p* value for comparison between the studied groups; *: statistically significant at *p* ≤ 0.05.

## Data Availability

Not applicable.

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
