# Peer review of "The Impact of SKP2 Gene Expression in Chronic Myeloid Leukemia"

_genes, 2022, doi:10.3390/genes13060948_

Round 1

Reviewer 1 Report

Hodeib and colleagues analyze the expression of SKP2 in CML in a cohort of CML chronic phase patients and healthy controls. They also study SKP2 expression according to the type of molecular response at 3 months after treatment.

Some points should be improved prior to publication:

  • English language should be overall improved: verb tenses should be reviewed and unified.
  • Fusion nomenclature should be updated according to the latest HGVS recommendations (BCR::ABL1)
  • In the introduction section this sentence is contradictory “The initial molecular response at 3 months after TKI therapy for CML patients has been confirmed by several groups as early response landmark to predict the outcome. Moreover, several studies reported that the rate of BCR-ABL1 reduction is of a critical prognostic and predictive value in CML patients initially treated with TKI. Unfortunately, there is a little information in the literature, predicting treatment response in CML patients and the available data remain insufficient and unclear”
  • In the Methods section, BCR::ABL1 quantitation kit should be specified.
  • Why do authors perform mononuclear cells isolation? It should be explained in the text. Clinical guidelines state that BCR::ABL1 quantitation should be performed in peripheral blood and mononuclear cell isolation is not required.
  • It would be interesting to analyze SKP2 expression according to BCR::ABL1 the presence or absence of resistance mutations
  • In Figure 1 there patients seem to segregate at 3 months after IM according to SKP2 expression. Is there any difference between these two subsets of patients?
  • BCR::ABL1 fusion is a pathognomonic feature of CML and the main molecular marker for disease follow up. Stating that SKP2 expression could improve CML diagnostics is a too strong sentence.

Author Response

We would like to thank the reviewer very much for his comments and critiques and we are pleased to respond to it.

Reviewer 2 Report

The manuscript titled “The impact of SKP2 Gene Expression in Chronic Myeloid Leukemia” describes that the authors want to use the SKP2 gene expression in the PBMCs to diagnose chronic myeloid leukemia patients at least 3 months earlier than normal. Overall, the manuscript is not very well prepared which reduces the manuscript quality. The followings are some concerns and comments have been pointed out that the authors may want to consider.

  1. Line 15: Please use italic p as it refers to a p-value. Check throughout the manuscript. The p here should be lower case.
  2. Line 17: A space is needed before the word “conclusion”. Check throughout the manuscript.
  3. Line 19: The keywords are missing.
  4. Line 22: What’s the meaning of “an annual incidence of 1-2 cases per 100,000/year”?
  5. Line 65 Methods section: 1) PBMCs isolation should be described, in brief at least. 2) Total RNA extraction from PBMCs should be described.
  6. Line 93: Define “IS%” before using it.
  7. Line 104: The authors already stated that GAPDH is the reference gene in this study, there is no need to describe it again on line 115.
  8. Line 114: The authors emphasize that SKP2 is the gene of interest. I’d suggest moving it to the beginning of this paragraph.
  9. Lines 109-113: Please separate each primer into an individual line to make it looks better.
  10. Line 115: The “−ΔΔCT” should be superscript. Delete spaces to make it looks better.
  11. Line 120: The “Student t-test” should be “Student’s t-test”.
  12. Line 121: The “Mann Whitney test” should be the “Mann-Whitney U test”.
  13. Line 128: The “Kruskal Wallis test” should be the “Kruskal-Wallis test”.
  14. Lines 133-175: The paragraph is too long. I’d suggest the authors split it into at least two paragraphs for easier tracking.
  15. Line 176 Table 1: The “NA” should be defined on lines 177-178.
  16. Line 181: What’s the meaning of “p1”?
  17. Line 184 Figure 1: The legend for Y-axis is missing. The statistical method should be briefly described in the figure legend; the significance should be labeled in the figure image.
  18. Line 185: Please be consistent with or without space before or after the equal sign throughout the manuscript.
  19. Line 191: Provide higher resolution Figure 2.
  20. SKP2: I highly suggest the authors provide SKP2 ELISA data to verify both gene level and protein level expression. It might be a good guideline for the possible further daily clinical practice.
  21. Line 286: Provide more details for the funding information.

Author Response

(The authors gave the same response as above.)

Round 2

Reviewer 1 Report

Please correct the fusions gene name (BCR::ABL  BCR::ABL1) throughout the text

Some sentences make no sense, for example:

-       Line 31:  It is noteworthy that TKI resistance either BCR::ABL-31 dependent or independent resistance

-       Line 33: overexpression BCR::ABL

-       Line 76: and plays integral role in the initiation and the progression of CML, modulating the SKP2/Bcr::Abl axis via stabilizing SKP2

-       Lines 199-200: In total, 100 patients recently diagnosed as CML and 100 healthy subjects (control 199 group). “ This sentence makes no sense

-        

Separate paragraph starting at line 41 (change of subject)

Lines 96-99: rewrite, verb tense does not match

The Kolmogorov- Smirnov was used (Line 184)  test missing

I still don’t understand why authors perform PBMC cell isolation instead of using whole blood for SKP2 expression analysis.

Author Response

We would like to thank you very much for your comments and valuable effort

  • Please correct the fusions gene name (BCR::ABL>>BCR::ABL1) throughout the text

Thanks for your comment

Modified as required throughout the text

  • Some sentences make no sense, for example:

-       Line 31:  It is noteworthy that TKI resistance either BCR::ABL-31 dependent or independent resistance

Thanks for your comment

Modified as required by the English Edit line 33, 34 in the introduction section in the revised version

-       Line 33: overexpression BCR::ABL

Thanks for your comment

Modified as required by the English Edit line 35, 36 , 37 in the introduction section in the revised version

-       Line 76: and plays integral role in the initiation and the progression of CML, modulating the SKP2/Bcr::Abl axis via stabilizing SKP2

Thanks for your comment

Modified as required by the English Edit and the author line 59, 60, 61 in the introduction section in the revised version

-       Lines 199-200: In total, 100 patients recently diagnosed as CML and 100 healthy subjects (control 199 group). “ This sentence makes no sense

Thanks for your comment

Modified as required by the English Edit line 183,184 in the result section in the revised version

-   Separate paragraph starting at line 41 (change of subject)

Thanks for your comment

Modified as required by the English Edit line 45 in the introduction section in the revised version

- Lines 96-99: rewrite, verb tense does not match

Thanks for your comment

Modified as required by the English Edit line 102 in the method section in the revised version 

- The Kolmogorov- Smirnov was used (Line 184)  test missing

Thanks for your comment

Modified as required by the English Edit line 165 in Statistical analysis section in the revised version 

- I still don’t understand why authors perform PBMC cell isolation instead of using whole blood for SKP2 expression analysis.

Thanks for your comment

Total RNA extraction from PBMC or whole blood is the same. We performed PBMC cell isolation (extra step and cost) to use a part of the sample in the present study and to store the remaining of the sample for future studies or extend the present study on the same patients.

Reviewer 2 Report

The following comments that the authors should consider and double-check to homogenous the format throughout the manuscript to improve the quality of the manuscript. Please provide a clean version after the manuscript has been revised. The location numbers for the line and paragraph in the response should be matched with your revised version for easier tracking.

1.       Moderate English changes still are required. For example, line 76 should be “plays an integral role”;  line 81 should be “an early time point”; and so on.

2.       Please seriously check throughout the manuscript “an extra space is needed”, or “an extra space should be deleted”.

3.       Line 21 section 1. Introduction: Please split the introduction part, instead of only one paragraph.

4.       Line 125: Please provide a brief protocol here for qRT-PCR.

5.       Line 145: A space is needed before “In brief”, and followed “In brief” should be a “,” instead of “;”.

6.       Lines 160-163: There are extra spaces followed by “5 primer(5’)”; line 163, an extra space followed by “(3’)”. Double-check throughout the manuscript.

7.       Line 166 section 2.5.: At least a brief protocol for ELISA should include.

8.       Line 227: A space is needed between the word “was” and “significantly”.

9.       Line 296 Figure 1: a) Provide higher resolution Figure 1; b) The meaning of “*” should include in the Figure 2 legend; c) I’d suggest the authors use “Relative mRNA expression…” as the Y-axis legend; d) The p in the image should be italic;

10.   Line 321 Figure 2: a) Provide higher resolution Figure 2; b) The meaning of “*” should include in the Figure 2 legend; c) I’d suggest the authors use “SKP2 protein levels…” as the Y-axis legend, and check throughout the manuscript to make it clearer; d) The p in the image should be italic.

11.   Line 368 section 4. Discussion: Please split the discussion part. Only one paragraph is too long.

12.   Space is needed: Line 373, before the word “additionally”, line 374, before the words “Of note”. Please check throughout the manuscript and provide a clean version after the manuscript has been revised.

Author Response

We would like to thank you very much for your valuable comments and we are pleased to provide a point to point response

The following comments that the authors should consider and double-check to homogenous the format throughout the manuscript to improve the quality of the manuscript. Please provide a clean version after the manuscript has been revised. The location numbers for the line and paragraph in the response should be matched with your revised version for easier tracking.

Thank you very much

A clean revised version after English and plagiarism editing is submitted

  1. Moderate English changes still are required. For example, line 76 should be “plays anintegral role”;  line 81 should be “an early time point”; and so on.

Thanks for your comment

Modified as required by the English Editing service and the authors, line 60, 65 in the introduction section in the revised version 

  1. Please seriously check throughout the manuscript “an extra space is needed”, or “an extra space should be deleted”.

Thanks for your comment

Modified as required by the English Editing service and the authors throughout the text

  1. Line 21 section 1. Introduction: Please split the introduction part, instead of only one paragraph.

Thanks for your comment

Modified as required in the introduction section in the revised version

  1. Line 125: Please provide a brief protocol here for qRT-PCR.

Thanks for your comment

Paragraphs added as required line 137-142  in the method section 2.4 in the revised version 

  1. Line 145: A space is needed before “In brief”, and followed “In brief” should be a “,” instead of “;”.

Thanks for your comment

Modified as required by the English Editing service and the authors throughout the manuscript in the revised version 

  1. Lines 160-163: There are extra spaces followed by “5 primer(5’)”; line 163, an extra space followed by “(3’)”. Double-check throughout the manuscript.

Thanks for your comment

Modified as required by the English Edit and the author throughout the manuscript in the revised version 

  1. Line 166 section 2.5.: At least a brief protocol for ELISA should include.

Thanks for your suggestion

Paragraphs were added as required lines 153-162 in the method section 2.5 in the revised version 

  1. Line 227: A space is needed between the word “was” and “significantly”.

Thanks for your comment

Modified as required by the English Editing service and the authors in the revised version 

  1. Line 296 Figure 1: a) Provide higher resolution Figure 1; b) The meaning of “*” should include in the Figure 2 legend; c) I’d suggest the authors use “Relative mRNA expression…” as the Y-axis legend; d) The p in the image should be italic;

Thanks for your suggestion

Figure 1 was deleted and the new figure was added as required and the legend was modified as required 

  1. Line 321 Figure 2: a) Provide higher resolution Figure 2; b) The meaning of “*” should include in the Figure 2 legend; c) I’d suggest the authors use “SKP2 protein levels…” as the Y-axis legend, and check throughout the manuscript to make it clearer; d) The p in the image should be italic.

Thanks for your suggestion

Figure 2 was deleted and the new figure was added as required and the legend was modified as required and SKP2 protein levels was modified as required throughout the manuscript

  1. Line 368 section 4. Discussion: Please split the discussion part. Only one paragraph is too long.

Thanks for your comment

Modified as required in the discussion section in the revised version 

  1. Space is needed: Line 373, before the word “additionally”, line 374, before the words “Of note”. Please check throughout the manuscript and provide a clean version after the manuscript has been revised.

Thanks for your comment

Modified as required by the English Editing service and the authors throughout the text

Best regards